# Hyperbaric Oxygen Preconditioning Upregulates Heme OxyGenase-1 and Anti-Apoptotic Bcl-2 Protein Expression in Spontaneously Hypertensive Rats with Induced Postischemic Acute Kidney Injury

**DOI:** 10.3390/ijms22031382

**Published:** 2021-01-30

**Authors:** Jelena Nesovic Ostojic, Milan Ivanov, Nevena Mihailovic-Stanojevic, Danijela Karanovic, Sanjin Kovacevic, Predrag Brkic, Maja Zivotic, Una Jovana Vajic, Djurdjica Jovovic, Rada Jeremic, Senka Ljubojevic-Holzer, Zoran Miloradovic

**Affiliations:** 1Department of Pathophysiology, Medical Faculty, University of Belgrade, 11000 Belgrade, Serbia; sanjin.kovacevic@med.bg.ac.rs; 2Institute for Medical Research, Department of Cardiovascular Physiology, University of Belgrade, 11129 Belgrade, Serbia; ivmilan@imi.bg.ac.rs (M.I.); nevena@imi.bg.ac.rs (N.M.-S.); danijela.karanovic@imi.bg.ac.rs (D.K.); unajovana@imi.bg.ac.rs (U.J.V.); djurdjica@imi.bg.ac.rs (D.J.); zokim@imi.bg.ac.rs (Z.M.); 3Department of Medical Physiology, Medical Faculty, University of Belgrade, 11000 Belgrade, Serbia; wubrkic@yahoo.com (P.B.); rdjeremic@yahoo.com (R.J.); 4Department of Pathology, Medical Faculty, University of Belgrade, 11000 Belgrade, Serbia; majajoker@gmail.com; 5Department of Cardiology, Medical University of Graz, A-8036 Graz, Austria; senka.ljubojevic@medunigraz.at

**Keywords:** acute kidney injury, hyperbaric oxygen preconditioning, heme oxygenase-1, Bax, Bcl-2, Wistar rats, spontaneously hypertensive rats

## Abstract

Renal ischemia and reperfusion (I/R) injury is the most common cause of acute kidney injury (AKI). Pathogenesis of postischemic AKI involves hemodynamic changes, oxidative stress, inflammation process, calcium ion overloading, apoptosis and necrosis. Up to date, therapeutic approaches to treat AKI are extremely limited. Thus, the aim of this study was to evaluate the effects of hyperbaric oxygen (HBO) preconditioning on citoprotective enzyme, heme oxygenase-1 (HO-1), pro-apoptotic Bax and anti-apoptotic Bcl-2 proteins expression, in postischemic AKI induced in normotensive Wistar and spontaneously hypertensive rats (SHR). The animals were randomly divided into six experimental groups: SHAM-operated Wistar rats (W-SHAM), Wistar rats with induced postischemic AKI (W-AKI) and Wistar group with HBO preconditioning before AKI induction (W-AKI + HBO). On the other hand, SHR rats were also divided into same three groups: SHR-SHAM, SHR-AKI and SHR-AKI + HBO. We demonstrated that HBO preconditioning upregulated HO-1 and anti-apoptotic Bcl-2 protein expression, in both Wistar and SH rats. In addition, HBO preconditioning improved glomerular filtration rate, supporting by significant increase in creatinine, urea and phosphate clearances in both rat strains. Considering our results, we can also say that even in hypertensive conditions, we can expect protective effects of HBO preconditioning in experimental model of AKI.

## 1. Introduction

Acute kidney injury (AKI) is manifested by a rapid decline in renal function, especially glomerular filtration rate (GFR), occurring over a few hours or days, followed by failure to maintain fluid, electrolyte and acid-base homeostasis [1]. Diabetes, hypertension and older age are risk factors for acute kidney injury [2]. AKI is also recognized as a potential in-hospital complication of sepsis, heart condition and surgery [2,3]. Renal ischemia and reperfusion (I/R) injury is the most common cause of AKI [4]. Pathogenesis of renal I/R injury involves hemodynamic changes, oxidative stress, inflammation process, calcium ion overloading, apoptosis and necrosis [4]. The relative contribution of each to decline GFR is uncertain [5]. Up to date, therapeutic approaches to treat AKI are extremely limited, so all efforts are made to prevention, early detection of the disorder and establishing secondary preventive measures to impede AKI progression [6].

Hyperbaric oxygen (HBO) therapy can be the part of the treatment for many patients suffering from diseases with underlying hypoxia [7]. This therapeutic modality is based upon administration of 100% molecular oxygen under the increased pressure, most frequently two times greater than the atmospheric pressure at sea level [8]. HBO preconditioning and therapy provide protective and beneficial effects in I/R injury, especially of brain and heart [9,10,11]. On the other hand, there are not much data about the effects of HBO preconditioning applied in experimental models of postischemic AKI, especially accompanied with hypertension [12]. Several mechanisms have been demonstrated, by which HBO therapy and preconditioning achieves ameliorate effects on I/R injury. These mechanisms include: improving oxygen delivery to an area with diminished blood flow [7], decreasing free oxygen radicals production and ameliorating the consequences of hypoxia and ischemia like tissue edema, increased permeability of blood vessels, derangement of tissue metabolism and inflammation [8,13]. In AKI, renal, urinary and plasma KIM-1 levels are significantly elevated in a short period of time and correspond to the extent of renal damage [14,15]. KIM-1 is not only the early marker of kidney injury, but also a predictive marker for future decline in kidney function and risk of chronic kidney disease [14,16]. Furthermore, KIM-1 is mainly expressed in differentiated proximal tubule epithelial cells, especially in proximal tubule S3 outer medulla area because of its sensitivity to ischemia, hypoxia and toxicity [14].

Heme oxygenase (HO) is a rate-limiting enzyme in the catabolism of heme [17]. It exists in two isoforms, HO-1, the inducible form, and HO-2, the constitutive form [18]. Both enzymes catalyze the degradation of heme to biliverdin, iron and carbon monoxide (CO) [18]. HO-2 is localized in the brain, testis as well as in the vascular endothelium [19]. It was shown that HO-1 protected the kidney in several models of AKI, including ischemia and glycerol-induced AKI, nephrotoxic serum nephritis, cisplatin nephrotoxicity and AKI induced by rapamycin [20,21,22,23,24]. Because of HO-1 cytoprotection, a spectrum of drugs has been used to up-regulate HO-1 expression and its activity [25]. In addition, heme oxygenase-1, is important in the regulation of cell proliferation, differentiation, oxidant/antioxidant systems, and apoptosis, thereby affecting inflammatory processes and immune response [26]. Earlier it had been showed that cytoprotective effect of HO-1 was associated with increased anti-apoptotic proteins Bcl-2 and Bcl-xl and a decrease of pro-apoptotic caspase-3 and caspase-9 along with increased expression of inactive BAX [17]. On the other hand, HBO may directly affect cell apoptosis, signal transduction and gene expression in those that are sensitive to oxygen or hypoxia. Translating the protective and beneficial effects of HBO into current practice requires an understanding of the mechanisms by which HBO achieves these positive effects in I/R injury, as well as considering risk factors, comorbidities, and the use of some medications that may interfere with HBO therapy [27]. Thus, the aim of this study was to evaluate the effects of HBO preconditioning on HO-1, pro-apoptotic Bax and anti-apoptotic Bcl-2 proteins expression, in postischemic acute kidney injury induced in normotensive Wistar and spontaneously hypertensive rats.

## 2. Results

### 2.1. Hemodynamic Parameters

Systolic and diastolic blood pressure (SAP and DAP), are shown in Table 1. Both in normotensive and hypertensive groups SAP and DAP were significantly decreased in AKI group compared to SHAM group. In groups with HBO preconditioning (W-AKI + HBO, SHR-AKI + HBO) no differences were observed compared to AKI groups.

### 2.2. Glomerular Filtration Rate

Glomerular filtration rate estimated by creatinine (C_Cr_), urea (C_U_) and phosphate (C_Phos_) clearances are shown in Table 2. AKI induction resulted in a significant reduction of C_Cr_, C_U_ and C_Phos_ when compared to SHAM group in both, normotensive and hypertensive rats. Remarkable improvements in C_Cr_, C_U_ and C_Phos_ were observed in groups with HBO preconditioning compared to AKI groups (both, normotensive and hypertensive).

### 2.3. Plasma KIM-1 Levels

KIM-1 levels were significantly increased in AKI group compared to SHAM operated rats in both, normotensive and hypertensive groups. Remarkable decrease in KIM-1 levels was noticed in SHR-AKI + HBO group (*p* < 0.05) compared to SHR-AKI group, thus this value become not significantly differ compared to SHAM (Figure 1).

### 2.4. Histological Findings

Mostly normal morphology of renal tissue, including glomerular and tubulointerstitial compartments were observed in the SHAM operated normotensive (Figure 2A) and hypertensive (Figure 2D) rats. In animals with AKI, both normotensive (Figure 2B) and hypertensive (Figure 2E) significant morphological alterations were present: proximal tubular dilatation, necrosis of tubular epithelial cells and PAS-positive casts. In treated animals (W-HBO + AKI and SHR-HBO + AKI) morphological changes were represented at less extent compared to their AKI control, with reduced tubular dilatation, necrosis and formation of PAS positive cast (Figure 2C,F).

### 2.5. Bax, Bcl-2 and HO-1 Expression in Kidney Tissue

The protein expressions of Bax, Bcl-2 and HO-1 are shown in Figure 3. In normotensive, Wistar groups, AKI induction significantly increased Bcl-2 and HO-1 expression, while it decreased Bax expression compared to SHAM operated rats. In opposite, HBO preconditioning significantly increased Bax expression compared to W-AKI group. On the other hand, in hypertensive rats, AKI induction decreased Bax expression. In group with HBO preconditioning before AKI induction, expression of Bax and HO-1 was significantly increased compared to AKI group without preconditioning.

## 3. Discussion

Previously, we have shown that HBO preconditioning succesfully blunted the harmful effects of IR injury in AKI of Wistar as well as SH rats by improving systemic oxidative stress, antioxidant defense, therefore normalized renal hemodynamic parameters [12,28]. Here we focused on the effects of HBO preconditioning on renal function through the assessment of GFR and its association with the endogenous antioxidant heme oxygenase-1 and markers of apoptosis, in the same experimental model. We demonstrated significant improvement of GFR (the increase in value of creatinine, urea and phosphate clearances) in AKI + HBO compared to AKI group in rats with and without hypertension. In addition, we demonstrated that HBO preconditioning upregulated HO-1 and anti-apoptotic Bcl-2 protein, in both Wistar and SH rats. Considering our previous and present results, we can also say that even in hypertensive conditions, we can expect protective effects of HBO preconditioning in experimental model of AKI.

We found significant decrease in systolic and diastolic arterial pressure (SAP and DAP), after the AKI induction in both strains of rats. These results are similar to data obtained by Bowmer et al., in experimental study of glycerol-induced AKI [29]. They assumed and discussed the importance of high uremia in diminishing α_1_ adrenoreceptors sensitivity, as a potential cause of SAP and DAP reduction. This also correlate to decreased creatinine, urea and phosphate clearances that we showed in this study. In our experimental model, HBO preconditioning did not affect SAP and DAP values, nor in Wistar, neither in SH rats, after AKI induction, that was in accordance with Rubinstein et al. study data [30].

Our results showed that HBO preconditioning significantly decreased plasma KIM-1 levels after postischemic AKI, in SH rats. In W-AKI + HBO group, KIM-1 values were also reduced compared with W-AKI group but not at significant manner. On the other hand, parameters of renal function, creatinine, urea and phosphate clearances, were significantly improved after HBO preconditioning, in both strains of rats, supporting the idea of ameliorating effect of this therapeutical modality in postischemic AKI. The major morphological changes in ischemic kidney, that can be seen by light microscopy include effacement and loss of proximal tubule brush border, patchy loss of tubular cells, focal areas of proximal tubule dilatation and distal tubule casts, and areas of necrosis [31]. By histopathological examination, we confirmed, that functional improvement in postischemic AKI, after HBO preconditioning was followed by ameliorate structural features of kidney tissue in rats with and without hypertension. These morphological changes included less sever lesions of tubular epithelial cells, reduced tubular dilatation and fewer PAS positive casts.

Besides, hypertension, diabetes and older age, there are some other risk factors for AKI, including renal artery stenosis, cerebrovascular disease, heart failure, coronary disease [2,32]. By recognizing these risk factors for the AKI development, we can find potential candidates for HBO preconditioning. Burlacu et al. in their study evaluated that almost 17% of consecutive patients with acute myocardial infarction (AMI) referred for primary percutaneous intervention from a single tertiary center, also presented renal artery stenosis (RAS). Moreover, they described a particular hydration, metabolic and endothelial profile of these RAS plus AMI patients [33]. As it is mentioned that renal artery stenosis was a risk factor for AKI [2,32], especially in condition when some cardiac surgery is needed, patients with these particular hydration, metabolic and endothelial profile might be a potential candidate for HBO preconditioning.

We still do not know all the pathways how HBO therapeutical modality ameliorates the renal function in experimental postischemic AKI, but we can assume that several mechanisms, are involved. In addition to improvement in renal blood flow and oxidative stress parameters [12,28], increased HO-1 expression with its cytoprotective action, as well as increased anti-apoptotic Bcl-2 protein can contribute to HBO preconditioning beneficial effects, in experimental model of AKI.

Heme oxygenase-1 (HO-1) is a homeostatic enzyme upregulated in stress [34]. In response to injury, the kidney is able to provoke adaptive and protective mechanisms to limit further damage [35]. One such mechanisms to limit further damage [21,36] is increased expression of HO-1 in the kidney. Studies have shown that HO-1 mRNA is induced in the kidney as early as 3 to 6 h in animal models of I/R and nephrotoxin induced AKI [21,22]. Zager et al. illustrate that plasma and urinary HO-1 concentrations are excellent biomarkers of intrarenal HO-1 gene activity during the initiation phase of AKI [37]. Our results are in accordance with these findings, and implicate increased HO-1 expression as one of the main protective mechanisms after postischemic AKI in Wistar rats. This is supported by the fact that HO-1 induced by oxidative stress plays a crucial role in protection against oxidative insult in diabetes and cardiovascular diseases [38].

In this study, we demonstrated, for the first time, that hyperbaric HBO preconditioning upregulates HO-1 expression in renal tissue, after AKI induced in SHR. He et al. obtained that HBO preconditioning induced tolerance against renal I/R injury via increased expression of HO-1 in normotensive Sprague-Dawley rats [39]. In our study, HO-1 expression was only mildly increased in AKI + HBO normotensive Wistar rats, probably due to previous compensatory increasing after AKI induction. HO-1, also called heat shock protein (HSP)32 is the first molecular which is reported to be a mediator for achieved improvement induced by HBO, particularly by increasing the tolerance of the organism against oxidative damage [40]. Rothfuss et al. demonstrated involvement of HO-1 in the adaptive protection of human lymphocytes after hyperbaric oxygen treatment [41]. Our results also showed no statistical difference in HO-1 expression between SHAM and SHR-AKI. We can explain these findings with observations obtained from different studies. The main pathophysiological properties of hypertension include enhanced vascular inflammation, blood vessels remodeling, increased vascular tone and enhanced oxidative stress [42]. Many physiological pathways are activated in an attempt to neutralize tissue damage provoked by different noxious stimuli, like inflammation and oxidative stress are. One of them is the heme oxygenase (HO) system. However, the pathophysiological activation of the HO system in hypertension could lead only to a transient increase in HO activity that is followed by the fall below the values that are capable to induce activation of different pathways that mediate HO-1 activity [42]. Therefore, a more intense potentiation of the HO system, for example by pharmacological agents such as hemin, cobalt protoporphyrin or through retroviral HO-1 gene delivery would be needed to overcome the threshold for cytoprotection. Hyperbaric oxygen preconditioning could play the potential role of a stronger stimulator of HO-1 system needed to increase HO-1 activity and its cytoprotective effect. This may be a possible explanation why we found no difference in HO-1 expression between SHAM and AKI group in SHR, but statistically significant increase in HO-1 expression in SHR-AKI after hyperbaric oxygen preconditioning.

This study showed that HBO preconditioning increased expression of the anti-apoptotic Bcl-2 protein, after induction of postischemic AKI in rats with and without hypertension. Such findings, are in part, similar to results obtained by Goodman et al. [17]. Further, we found that the increase of kidney Bcl-2 have been followed with an elevation of kidney HO-1 expression in both strains of rats preconditioned with HBO before AKI. So, we can assume that this increase in anti-apoptotic Bcl-2 proteins may be provoked at least in part, by an increase in HO-1 activity. In consistent with our results, Sheng et al. reported that HO-1 could reduce and inhibit apoptosis of lung tissue during cardio-pulmonary bypass in rats, and its anti-apoptotic effect may be achieved by upregulating the expression of anti-apoptotic protein bcl-2 in lung tissue [43]. Several other studies also confirmed upregulation of Bcl-2 [44,45,46] and heme oxygenase [47] in renal I/R injury model. Overexpression of Bcl-2 can prevent release of mitochondrial cytochrome c thereby suppressing apoptosis in renal tubule cells induced by hypoxia/reoxygenation [48].

On the other hand, our study found that preconditioning with HBO normalizes pro-apoptotic Bax protein expression in both Wistar and SH rats, subjected to AKI. Possible reason for this finding, may be very subtle interaction of Bax and Bcl-2 proteins, but also the fact that during the time after the ischemic insult, apoptosis and necrosis distributed in proximal and distal tubules are represented at different extent. Boventure et al., also showed on pathologic examination of the postischemic kidney, that the necrotic cell death of proximal tubules is initially more prominent than apoptotic cell death [49]. Distal tubule segments, including medullary thick ascending limb, are more glycolytic and less hypoxia sensitive than proximal tubule [50]. This contributes to their resistance to necrosis and instead favors development of apoptosis [51,52], or sublethal injury. Gobe et al. also showed that distal tubular epithelial cells are resistant to ischemia-induced injury via adaptive Bcl-2/Bcl-X_L_ upregulation, that is followed by minimal cell death by apoptosis [44]. They also concluded that these distal tubules function as a reservoir for the production of growth factors critical to its own maintenance and/or regeneration and also to preservation of the proximal tubule that abuts them, with only minimal apoptosis occurring. Bax expression is reported to be downregulated after some instances of ischemic renal damage, and is related to tubular regeneration [45]. Furthermore, apoptosis becomes increasingly important over time after the initiating insult [44,46,53]. Zhang et al. also showed that in experimental unilateral obstruction, 3 days after the experiment, the Bax was initially low, though increased with time as the Bcl-2 level fell [54]. In presented study, we demonstrated that hyperbaric oxygen preconditioning increased pro-apoptotic Bax as well as anti-apoptotic Bcl-2 protein, measured 24 h after ischemic insult in rats with and without hypertension. Damaged tubules expressing both Bcl-2 and Bax possibly could be explained by a self-protection mechanism, that is, an original Bax-positive damaged cell may struggle to survive by up-regulation of Bcl-2. It could also be explained by the exhaustion of this self-protection mechanism, that is, a Bcl-2 positive cell persistently exposed to damaging factors may finally express Bax and undergo apoptosis. Possible reason for, somewhat surprisingly high values of Bax in SHAM groups of Wistar and SH rats, compared to W-AKI, and SHR-AKI group could be the fact that during SHAM surgery, one kidney was removed, that possible might provoke increased apoptosis.

On the other hand, decreased Bax expression, implied decreased apoptosis in W-AKI and SHR-AKI might be explained by the severity of the ischemic lesions. A mild form of the ischemic insult can lead to apoptosis, while a severe form can lead to necrosis [55].

Taking together our results and results from the other studies, we can agree with Ueda et al. that the regulation of anti-apoptotic and pro-apoptotic proteins may vary with cell types in the kidney and that this may account for occurrence of apoptosis and susceptibility to apoptotic stimuli during the acute phase of acute kidney injury as well as the regeneration process during the recovery phase of the disease [56].

## 4. Materials and Methods

### 4.1. Ethics Statement

The experimental protocol was approved by the Ethic Committee of the Institute for Medical Research, University of Belgrade, and by the Veterinary Directorate, Ministry of Agriculture and Environmental Protection, Republic of Serbia (323-07-02449/2014-05) according to the National Law on Animal Welfare (‘Službeni Glasnik’ No. 41/09, 39/10) that is consistent with guidelines for animal research and principles of the European Convention for the Protection of Vertebrate Animals Used for Experimental and Other Purposes (Official Daily N. L 358/1-358/6, 18, December 1986) and Directive on the protection of animals used for scientific purposes (Directive 2010/63/EU of the European Parliament and of the Council, 22 September 2010).

### 4.2. Animals

In this study we used male spontaneously hypertensive rats (SHR, descendants of breeders originally obtained through Taconic Farms, Germantown, NY, USA) and their normotensive control, Wistar rats 24 weeks old and about 300 g weights. The animals were bred at the Institute for Medical Research, University of Belgrade, Serbia, and kept under controlled laboratory conditions (constant temperature 22  ±  1 °C, humidity of 65  ±  1%, 12 h light/dark cycle). The animals were divided in groups of four rats per cage and fed with a standard chow for laboratory rats (Veterinarski zavod, Subotica, Serbia). They were allowed free access to food and water. During the whole study, all experimental animals were monitored at least once per day, including weekends and public holidays.

### 4.3. Experimental Design

In SHR rats, hypertension was confirmed by indirect measurement on tail artery (Narco Bio Systems Inc., Houston, TX, USA). The animals were randomly divided into six experimental groups: sham-operated Wistar rats (W-SHAM, *n* = 8), Wistar rats with induced postischemic AKI (W-AKI, *n* = 7) and Wistar group with HBO preconditioning before AKI inducing (W-AKI + HBO, *n* = 8). On the other hand, SHR rats were also divided into same three groups: SHR-SHAM (*n* = 8), SHR-AKI (*n* = 8) and SHR-AKI + HBO (*n* = 8).

The rats of both strains were undergone to HBO preconditioning by placing the animals into custom made experimental HBO chamber (Holywell Neopren, Belgrade, Serbia), where they were exposed to 100% oxygen according to the following protocol: 10 min of slow compression, 2.0 atmospheres absolute (ATA) for 60 min, and 10 min of slow decompression, twice a day, at 12 h interval, during two-day period. The last HBO exposition was performed 24 h before AKI induction. When the desired pressure is reached, the flow of oxygen was reduced to maintain constant pressure that allows the flow out of the chamber. A reduction in CO_2_ accumulation in the chamber environment has been achieved by this constant exchange accompanied by a tray of calcium carbonate crystals. This protocol corresponds to a standard hyperbaric oxygen treatment that is routinely used in the clinical setting of Center for Hyperbaric Medicine, Belgrade, Serbia [57], and is in accordance with recommendations of The Committee of the Undersea and Hyperbaric Medical Society [58]. Each exposure was started at the same hour to exclude any confounding issues associated with the changes in biological rhythm. Body temperature was measured after the HBO preconditioning and no significant changes were found. In animals, AKI was induced 12 h after the last HBO preconditioning.

During all surgical procedures, the rats were anaesthetized by injecting 35 mg/kg body weight (b.w.) sodium pentobarbital intraperitoneally. AKI was induced by removal of the right kidney and atraumatic clamp occlusion of the left renal artery for 45 min. In SHAM-operated group the same surgical procedure was performed, but without left renal artery clamping. At the end of AKI induction, the abdominal incision was sutured and rats were placed into metabolic cages for 24 h, in order to collect urine samples, with free access to food and water. In order to alleviate postoperative pain, ketoprofen (5 mg/kg b.w.) was administrated subcutaneously.

### 4.4. Hemodynamic Measurements

Systolic arterial pressure (SAP) and diastolic arterial pressure (DAP) were measured 24 h after reperfusion by direct method, through a femoral artery catheter (PE-50, Clay-Adams Parsippany, NY, USA), connected to a physiological data acquisition system (9800 TCR Cardiomax II-TCR, Columbus, OH, USA), as previously described [28]. During this procedure, the rats were anaesthetized by injecting 35 mg/kg b.w. sodium pentobarbital intraperitoneally.

### 4.5. Sample Collection

Urine samples were collected before hemodynamic measurements, while blood samples, obtained by puncture of the abdominal aorta, were collected after hemodynamic measurements in the terminal phase of the study. They were collected into tubes containing lithium-heparin (Li-heparin, Sigma-Aldrich, St. Louis, MO, USA) and used for further analysis. Blood was centrifuged to separate plasma from erythrocytes. Until assaying, plasma samples were stored at −20 °C and erythrocytes samples at −80 °C. After blood samples collection, animals were sacrificed by pentobarbital overdose injection. For determination of morphological changes, kidney tissue was removed immediately after sacrificing and then prepared for histological examination.

### 4.6. Glomerular Filtration Rate

All biochemical parameters for the estimation of glomerular filtration rate (GFR), as a marker of kidney function, were measured using the automatic COBAS INTEGRA 400 plus (Hoffmann-La Roche, Germany) analyzer. Creatinine (C_Cr_), urea (C_U_) and phosphate (C_Phos_) clearances were calculated according to standard formula and normalized to body weight.

### 4.7. Determination of Plasma Kidney Injury Molecule-1 Levels

Plasma kidney injury molecule-1 (KIM-1) was determined by sandwich enzyme-linked immunosorbent assay (ELISA) kit according to manufacture instructions (R&D Systems, Inc., Minneapolis, MN, USA). The detection range for KIM-1 was 7.8–500 pg/mL.

### 4.8. Western Blot Analyses

The renal tissues were homogenized in lysis buffer, as previously described [59]. Equal amounts of protein samples (concentration determined by BCA assay, Thermo Fisher scientific, USA) were separated by 12% SDS-PAGE and transferred to nitrocellulose membrane (AppliChem, Darmstadt, Germany). Next, membranes were blocked with 5% nonfat milk in TBS-Tween-20 (Serva, Heidelberg, Germany) for 1 h and probed with appropriate dilutions of primary antibodies for Bax (1:1000, 04-434 Millipore, Burlington, MA, USA), Bcl-2 (1:250, 05-729 Millipore), HO-1 (1:250, ab13248 Abcam, Cambridge, UK), and actin (1:500, A5060 Sigma-Aldrich) overnight at 4 °C. Peroxidase-conjugated goat anti-rabbit immunoglobulin (1:40,000, A0545 Sigma-Aldrich) or goat anti-mouse immunoglobulin (1:20,000, A5278 Sigma-Aldrich) were used as secondary antibody. Enhanced chemiluminescence (ECL) method (Serva, Heidelberg, Germany) and quantified using Image Lab software (Bio-Rad) were used for protein bands visualization.

### 4.9. Histological Examination

For histological observation, the renal tissue was fixed in 10% buffered formalin solution. Later, the kidney was dehydrated in alcohol and embedded in paraffin block, cut into 5 μm thick sections and stained by periodic acid-Schiff (PAS) reagent. By light microscopy, slides were examined by two independent pathologists blinded to the experimental protocol.

### 4.10. Statistical Analysis

Data are presented as the mean ± standard error of the mean (SEM). Statistical analysis was carried out using Student’s t-test for independent samples. First, in order to evaluate AKI induction, in both normotensive and hypertensive rats, AKI groups were compared to SHAM groups, and after, to evaluate the effects of HBO preconditioning, AKI groups were compared to AKI + HBO groups. *p* value <0.05 was considered significant. Statistical calculations were performed using GraphPad Prism for Windows (Version 7.0, GraphPad Software, La Jolla, CA, USA).

## 5. Conclusions

In this study, we showed, for the first time, that hyperbaric oxygen preconditioning upregulates heme oxygenase-1 and anti-apoptotic Bcl-2 protein expression in postischemic acute kidney injury induced in spontaneously hypertensive rats. Moreover, we demonstrated beneficial effect of HBO preconditioning, on GFR, supporting by improved creatinine, urea and phosphate clearances, in Wistar and SH rats. All these results were in accordance with improved structure confirmed by histological examination. Considering our results, we can also say that even in hypertensive conditions, we can expect protective effects of HBO preconditioning in experimental model of postischemic acute kidney injury. At the end, we can concluded that, hyperbaric oxygen preconditioning applied in experimental model of postischemic acute kidney injury offers promising results for further experimental investigations, with the hope that one day it may become a therapeutical modality in practice.

## Figures and Tables

**Figure 1 ijms-22-01382-f001:**
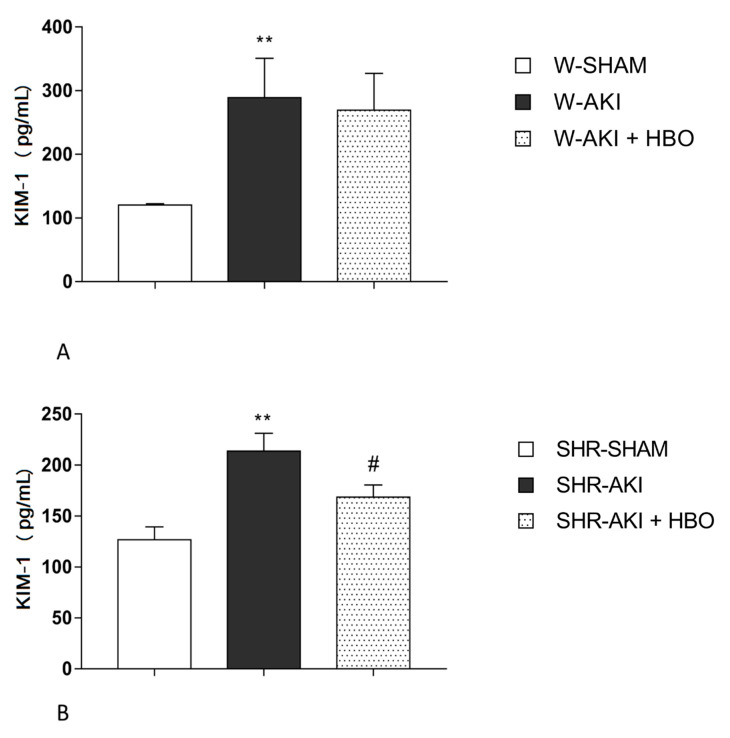
Plasma KIM-1 levels 24 h after reperfusion in normotensive (**A**) and hypertensive (**B**) groups. ** *p* < 0.01 vs. SHAM, # *p* < 0.05 vs. AKI.

**Figure 2 ijms-22-01382-f002:**
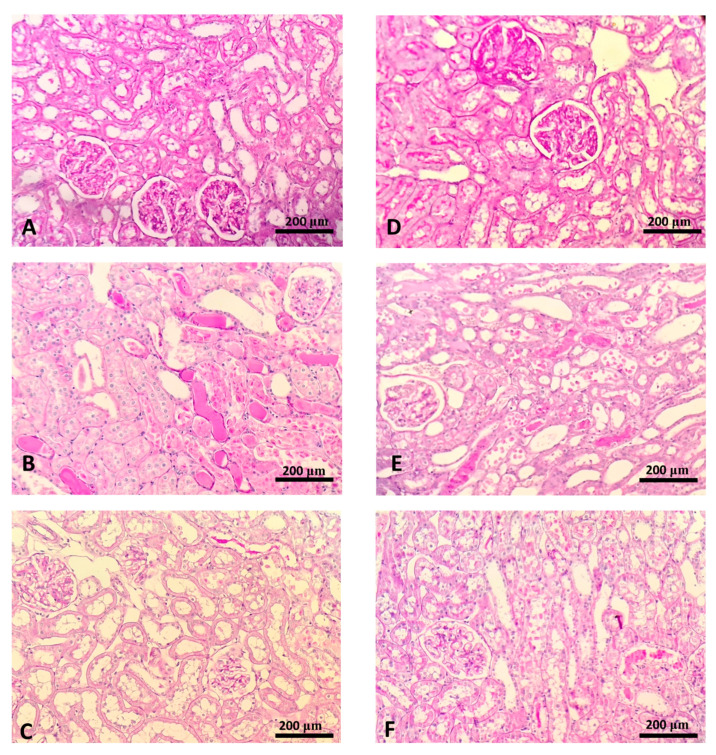
Histopathology of the representative kidney samples collected in different experimental groups (PAS staining, × 20 magnification): W-SHAM (**A**), W-AKI (**B**), W-HBO + AKI (**C**), SHR-SHAM (**D**), SHR-AKI (**E**), SHR-HBO + AKI (**F**).

**Figure 3 ijms-22-01382-f003:**
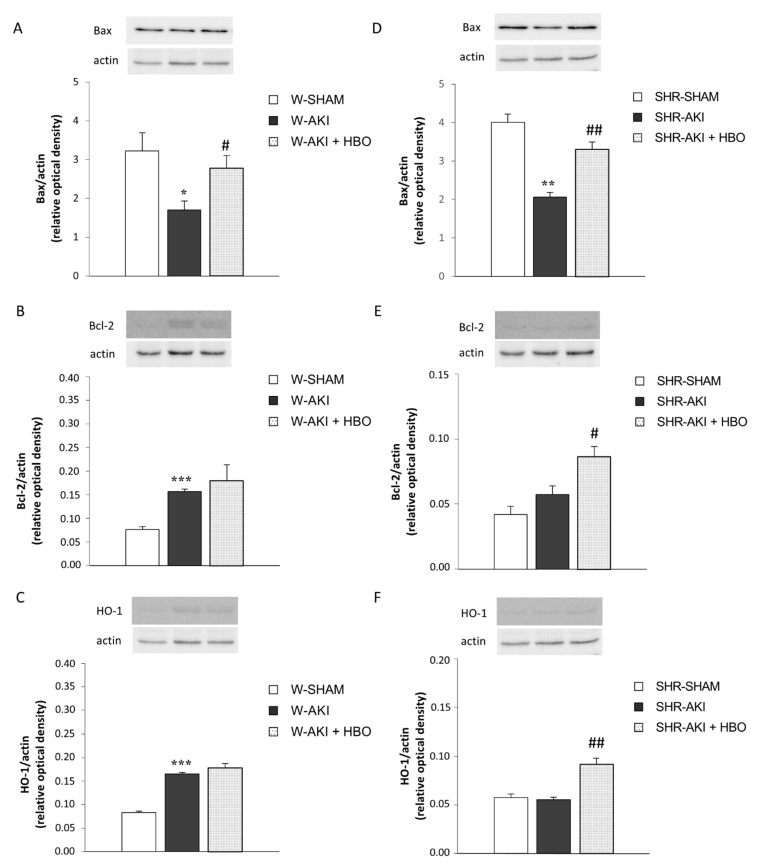
Bax, Bcl-2 and HO-1 expression in kidney tissue in normotensive (**A**–**C**) and hypertensive (**D**–**F**) groups. *** *p* < 0.001, ** *p* < 0.01, * *p* < 0.05 vs. SHAM, ## *p* < 0.01, # *p* < 0.05 vs. AKI.

**Table 1 ijms-22-01382-t001:** Systolic and diastolic blood pressure.

	SAP (mmHg)	DAP (mmHg)
W-SHAM	133.50 ± 5.80	79.25 ± 3.40
W-AKI	113.86 ± 3.99 **	70.14 ± 2.32 *
W-AKI + HBO	119.37 ± 6.48	69.62 ± 3.48
SHR-SHAM	167.62 ± 5.22	113.00 ± 6.54
SHR-AKI	138.25 ± 6.46 **	81.75 ± 10.49 *
SHR-AKI + HBO	131.50 ± 8.10	71.75 ± 4.83

SAP—systolic arterial pressure, DAP—diastolic arterial pressure, MAP—mean arterial pressure, * *p* < 0.05, ** *p* < 0.01 vs. SHAM groups.

**Table 2 ijms-22-01382-t002:** Creatinine (C_Cr_), urea (C_U_) and phosphate (C_Phos_) clearance 24 h after reperfusion.

	C_Cr_ (mL/min/kg)	C_u_ (mL/min/kg)	C_Phos_ (mL/min/kg)
W-SHAM	8.19 ± 0.35	3.94 ± 0.15	0.77 ± 0.09
W-AKI	0.51 ± 0.14 ***	0.21 ± 0.07 ***	0.35 ± 0.10 **
W-AKI + HBO	1.69 ± 0.54 ^#^	0.67 ± 0.22 ^#^	0.84 ± 0.21 ^#^
SHR-SHAM	4.03 ± 0.60	1.88 ± 0.32	0.71 ± 0.14
SHR-AKI	0.24 ± 0.02 ***	0.09 ± 0.01 ^***^	0.20 ± 0.03 **
SHR-AKI + HBO	1.22 ± 0.22 ^###^	0.21 ± 0.04 ^##^	0.43 ± 0.11 ^#^

C_cr_—creatinine clearance, C_U_—urea clearance, C_Phos_—phosphate clearance, ** *p* < 0.01, *** *p* < 0.001 vs. SHAM groups; # *p* < 0.05, ## *p* < 0.01, ### *p* < 0.001 vs. AKI groups.

## Data Availability

The data presented in this study are available in article.

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
