# Peer review of "Hyperbaric Oxygen Preconditioning Upregulates Heme OxyGenase-1 and Anti-Apoptotic Bcl-2 Protein Expression in Spontaneously Hypertensive Rats with Induced Postischemic Acute Kidney Injury"

_ijms, 2021, doi:10.3390/ijms22031382_

Round 1
Reviewer 1 Report
Thank you very much for the invitation to peer-review this article.
It is an article with a reasonable quality of English and has all the chapters of a well done preclinical research.
The Methodology is quite well presented and envisioned, as well as the presentation of the Results.
I believe that at the Discussion section, one can see improvements.
I think my main dissatisfaction is why the Authors recall on the issue of high blood pressure (one sentence in the abstract - I would suggest it to be removed; and a paragraph on Discussions - and I do not really understand what it has to do with the idea of ischemic preconditioning).
I realise that the Authors have good expertise in the field (with previous papers which were also quoted in this Manuscript), but they should stick to the issue of preconditioning or maybe to discuss more on renal artery stenosis implications in preconditioning (and multi-sit atherosclerosis: they can comment on this issue published in JAHA https://www.ahajournals.org/doi/10.1161/JAHA.115.002379?url_ver=Z39.88-2003&rfr_id=ori:rid:crossref.org&rfr_dat=cr_pub%20%200pubmed)
Overall, my conclusion: major changes requested for Acceptance.
Author Response
Response to Reviewer 1 Comments
Point 1: It is an article with a reasonable quality of English and has all the chapters of a well done preclinical research. The Methodology is quite well presented and envisioned, as well as the presentation of the Results.
Response 1: Thank you for the complements for our study and very useful comments for the manuscript.
Point 2: I believe that at the Discussion section, one can see improvements. I think my main dissatisfaction is why the Authors recall on the issue of high blood pressure (one sentence in the abstact - I would suggest it to be removed; and a paragraph on Discussions - and I do not really understand what it has to do with the idea of ischemic preconditioning).
Response 2: As postischemic acute kidney injury (AKI) is one of the most common form of AKI, our experiments were performed on animals subjected to postischemic AKI. Also,
there are many conditions that are risk factors for AKI development, including hypertension, oldery age, diabetes, coronary disease, renal artery stenosis, cardiac surgery...Because of a high prevalence of hypertension in the world population, our intention was to evaluate if protective effects of HBO preconditioning could be achieved even in hypertensive conditions (as hypertension is one of the most common disease in population, and risk factor itself to AKI). That is the reason, why we used spontaneously hypertensive rats. As we think that this is important part of our study, we did not remove the sentence from Abstract nor a paragraph from Discussion, but we rewrote this part of the text (line 31-33; 153-155). We hope that our explanation will be appropriate for the reviewer.
Point 3: I realise that the Authors have good expertise in the field (with previous papers
Which were also quoted in this Manuscript), but they should stick to the issue of
preconditioning or maybe to discuss more on renal artery stenosis implications in preconditioning (and multi-sit atherosclerosis: they can comment on this issue published
in JAHA https://www.ahajournals.org/doi/10.1161/JAHA.115.002379?url_ver=Z39.88-2003&rfr_id=ori:rid:crossref.org&rfr_dat=cr_pub%20%200pubmed)
Response 3: We found this issue published in JAHA https://www.ahajournals.org/doi/10.1161/JAHA.115.002379?url_ver=Z39.88-2003&rfr_id=ori:rid:crossref.org&rfr_dat=cr_pub%20%200pubmed very interesting. As renal artery stenosis is also the risk factor for AKI, we commented our results on this
issue and added a new paragraph in section Discussion (line 179-190).
Sincerely,
Jelena Nesovic Ostojic
Reviewer 2 Report
Results: The statistical test used for analysis of data is not appropriate for the study design. Please make new analysis- I suggest Two-way ANOVA. That way you will take into account the strain differences and the treatment effects, which is important for all of your conclusions.
Discussion: "...Considering our previous and present results, we can also assume that hypertension is not a risk factor for HBO preconditioning in experimental
model of AKI..." please clarify where did you in text described results that support this sentence and how did you asses if hypertension is a risk factor for HBO preconditioning.
Methods: please provide more information on blood pressure measurements- where they performed in anesthetized rats, and what kind of direct method is used?
Statistical analysis- the selected method Student t-test is not appropriate method for this study design. The results should have been compared with Two-way ANOVA and appropriate post hoc tests. Because of that, results obtained need to be revised and also conclusions need to be rewritten according to new statistics.
English- I would suggest English editing, due to a number of stylistic and typo errors, for better flow of the manuscript and easier reading.
Author Response
Response to Reviewer 2 Comments
Point 1: Results: The statistical test used for analysis of data is not appropriate for the study design. Please make new analysis- I suggest Two-way ANOVA. That way you will take into account the strain differences and the treatment effects, which is important for all ofyour conclusions.
Response 1: We appreciate reviewer comment and suggestion to make new analysis, Two-way ANOVA, that will take into account the strain differences. But our primary intention
was not to compare the effects of HBO preconditioning in postischemic AKI induced in two different rat strains, rather to observe certain parameters after HBO pretreatment in Wistar
and SH rats, separately. The reason why we used SH rats, besides Wistar, is high prevalence of hypertension in population and the fact that hypertension itself is a risk factor for AKI. So, by using SH rats, we wanted to evaluate if beneficial effects of HBO preconditioning could be achieved in experimental animals with hypertension. Our idea and primary concept were to evaluate the effects of HBO preconditioning (by observing clearances of creatinine, urea, and phosphate, KIM-1 and morphological characteristics of kidney tissue) in postischemic AKI induced in two different rat strains, but we did not compare Wistar and SHR with each other. As we confirmed beneficial effects of this pretreatment, we tried to explain potential mechanisams (by evaluating HO-1, Bax and Bcl-2 protein expression) that are involved in this protective effects. If we perform Two-way ANOVA, to compare two different strains, we would get more results, that would focused discussion to the other point of view. But we find that is a very good idea comparing the effects of HBO preconditioning in induced AKI in Wistar and SHR, and could be the subject for some further investigation.
Point 2: Discussion: "...Considering our previous and present results, we can also assume that hypertension is not a risk factor for HBO preconditioning in experimental model of AKI..." please clarify where did you in text described results that support this sentence and how did you asses if hypertension is a risk factor for HBO preconditioning.
Response 2: We agree with the reviewer that our observation "...Considering our previous and present results, we can also assume that hypertension is not a risk factor for HBO preconditioning in experimental model of AKI..." sounds inadequate, since we did not compare the parameters after HBO preconditioning between two different strains of rats (Wistar and spontaneously hypertensive), instead we only presented the results obtained in both strains. As we confirmed beneficial effects of HBO conditioning (by improving clearance of creatinine, urea and phosphate, KIM-1 and structural kidney changes) in postischemic AKI in SH rats, maybe it is better to say that even in hypertensive conditions, we can expect protective effects of HBO preconditioning in experimental model of postischemic AKI. We made this change in revised manuscript (Abstract, line 31-33 , Section Discussion, line 153-155, and in Conclusion line 409-411).
Point 3: Methods: please provide more information on blood pressure measurements-
where they performed in anesthetized rats, and what kind of direct method is used?
Response 3: Blood pressure were measured, in anesthetized rats, 24 hours after reperfusion, by a direct method, through a femoral artery catheter (PE-50, Clay-Adams Parsippany, NY, USA), connected to a physiological data acquisition system (9800TCR Cardiomax III-TCR, Columbus, OH, USA). We incorporated this part of text in the revised manuscript (line 348-354).
Point 4: Statistical analysis- the selected method Student t-test is not appropriate method for this study design. The results should have been compared with Two-way ANOVA
and appropriate post hoc tests. Because of that, results obtained need to be revised and also conclusions need to be rewritten according to new statistics.
Response 4: In this experimental study we evaluated the effects of HBO preconditioning in
postishemic AKI induced in two rat strains, separately. We decided, in consultation, with our
statistician to use Student t-test to compare Wistar SHAM and Wistar AKI, as SHAM was control in this experimental design, that served only to verify if postishemic AKI was
induced in experimental animals. When we showed the statistical difference in observed
parameters between these two groups, we confirmed that we performed a proper
experimental model. Then, we used Student t-test to compare Wistar AKI with Wistar AKI+ HBO. We made the same thing with SH rats, but we did not compare Wistar and SHR
with each other. By comparing SHAM groups of different strains, a statistically significant difference can be obtained between animals, even when they were not subjected by postischemic AKI, and this could lead to misinterpretation of the results.
Point 5: English- I would suggest English editing, due to a number of stylistic and typo errors, for better flow of the manuscript and easier reading.
Response 5: We went through the whole manuscript again, and corrected some stylistic and typo errors (line 107, 108; 156,157; 163,164; 169; 193-196; 226.229; 241,242; 272). We hope that this improve flow of the manuscript and make easier reading.
Sincerely,
Jelena Nesovic Ostojic
Round 2
Reviewer 1 Report
I believe that the Authors answered the points raised in the previous revision. I have no further comments. Congrats!
Reviewer 2 Report
I have no further comments on the manuscript, authors successfully replied.